# Some New Quantum BCH Codes over Finite Fields

**DOI:** 10.3390/e23060712

**Published:** 2021-06-03

**Authors:** Lijuan Xing, Zhuo Li

**Affiliations:** The State Key Laboratory of Integrated Services Networks, Xidian University, Xi’an 710071, China; ljxing@mail.xidian.edu.cn

**Keywords:** quantum stabilizer codes, BCH codes, cyclotomic cosets, dual codes

## Abstract

Quantum error correcting codes (QECCs) play an important role in preventing quantum information decoherence. Good quantum stabilizer codes were constructed by classical error correcting codes. In this paper, Bose–Chaudhuri–Hocquenghem (BCH) codes over finite fields are used to construct quantum codes. First, we try to find such classical BCH codes, which contain their dual codes, by studying the suitable cyclotomic cosets. Then, we construct nonbinary quantum BCH codes with given parameter sets. Finally, a new family of quantum BCH codes can be realized by Steane’s enlargement of nonbinary Calderbank-Shor-Steane (CSS) construction and Hermitian construction. We have proven that the cyclotomic cosets are good tools to study quantum BCH codes. The defining sets contain the highest numbers of consecutive integers. Compared with the results in the references, the new quantum BCH codes have better code parameters without restrictions and better lower bounds on minimum distances. What is more, the new quantum codes can be constructed over any finite fields, which enlarges the range of quantum BCH codes.

## 1. Introduction

QECCs are important tools to prevent quantum information from decoherence in quantum computations and quantum communications. After the fundamental research for QECCs [1,2,3], more and more good results have been proposed to improve the quantum codes.

There were relationships between quantum codes and classical self-orthogonal codes over finite fields [4,5,6]. The construction of binary quantum BCH codes was based on classical additive codes over GF(4) [4]. The conclusions in [4] could be generalized to all the nonbinary primitive quantum BCH codes over finite fields [7]. Aly et al. extended Steane’s results [8] to narrow-sense (not necessarily primitive) BCH codes with certain distances over GF(q) [5]. Nonbinary quantum codes with better code parameters were obtained by CSS construction [9]. Steane’s enlargement construction was generalized from binary quantum codes to *q*-ary quantum codes [10]. Moreover, two families of nonbinary quantum codes were presented by the Hermitian construction [11]. Some quantum codes could be constructed by negacyclic codes [12,13] and constacyclic codes [14,15]. Good nonbinary quantum codes were constructed by corresponding cyclotomic cosets with given parameters [16]. The designed quantum BCH codes were obtained with given code lengths [9,16,17,18,19,20].

However, quantum coding theory is aimed at finding codes with given parameter sets and optimizing the code parameters. The construction of quantum BCH codes is studied in this paper. First, we try to find such classical BCH codes which contain their dual codes by studying the suitable cyclotomic cosets. The suitable cyclotomic cosets are proven to have the highest numbers of consecutive integers in defining sets and compute the dimensions of quantum BCH codes correctly. Then, we can construct nonbinary quantum BCH codes with given parameter sets. Finally, a new family of quantum BCH codes can be realized by Steane’s enlargement of nonbinary Calderbank-Shor-Steane (CSS) codes and Hermitian construction.

This paper is organized as follows. The basic theory of classical BCH codes is defined in Section 2. New families of quantum BCH codes by Steane’s enlargement of CSS construction are constructed in Section 3. New families of quantum BCH codes by Hermitian construction generated by classical BCH codes over $F_{q^{2}}$ are shown in Section 4. The results are compared with corresponding references in Section 5.

## 2. Preliminaries

The finite field is denoted by $F_{q}$ with q elements, where q is a prime power. A linear code of length n over $F_{q}$ is a subspace of $F_{q}^{n}$.

**Definition** **1.**
*Given two vectors*
$x,y\in F_{q}^{n}$
*, the Euclidean inner product over*
$F_{q}$
*is defined as follows:*
${\langle x,y\rangle }_{E}=x_{0}y_{0}+x_{1}y_{1}+\ldots +x_{n-1}y_{n-1}.$


Similarly, given two vectors $x,y\in F_{q^{2}}^{n}$, the Hermitian inner product over $F_{q^{2}}$ is defined as follows: ${\langle x,y\rangle }_{H}=x_{0}y_{0}^{q}+x_{1}y_{1}^{q}+\ldots +x_{n-1}y_{n-1}^{q}.$

We define $\gcd\left(n,q\right)=1$ in this paper. The smallest positive integer $m_{0}$ in $q^{m_{0}}\equiv 1\;\mathrm{mod}n$ is called the multiplicative order of q modulo n and is denoted by $m_{0}=ord_{n}(q)$. Namely, $n|q^{m_{0}}-1$ holds.

If C is an $\left[n,k,d\right]$ code over $F_{q}$, the Euclidean dual code of C is defined as follows: $C^{⊥E}=\left\{x\in F_{q}^{n}|{\langle x,y\rangle }_{E}=0\;\mathrm{for}\;\mathrm{all}\;y\in C\right\}.$

If C is an $\left[n,k,d\right]$ code over $F_{q^{2}}$, the Hermitian dual code of C is defined as follows: $C^{⊥H}=\left\{x\in F_{q^{2}}^{n}|{\langle x,y\rangle }_{H}=0\;\mathrm{for}\;\mathrm{all}\;y\in C\right\}.$

The classic BCH code is a family of well-studied cyclic codes. Many explicit constructions of classical BCH codes [21] and QECCs [5] have been proposed so far. They can all be characterized by the cyclotomic cosets. Let φ[i]={iqz mod n|z∈ℤ} denote the *q*-ary cyclotomic coset of i modulo n.

**Definition** **2.**
*A BCH code*
C
*over*
$F_{q}$
*with length n and designed distance*
δ
*is a cyclic code. The defining set is denoted by*
$Ζ=\overset{b+\delta -2}{\underset{i=b}{U}}\varphi [i]$
*. If*
$n=q^{m_{0}}-1$
*, it is called a primitive BCH code. If*
b=1
*, it is called a narrow-sense BCH code.*


The minimal polynomial over $F_{q}$ of β is the lowest degree monic polynomial M(x), with coefficients from $F_{q}$ such that $M\left(\beta \right)=0$. If $\beta =\alpha^{i}$ for a fixed primitive *n*-th root of unity $\alpha \in F_{q^{m_{0}}}$, then the minimal polynomial of β over $F_{q}$ is denoted by $M^{\left(i\right)}(x)=$$\prod_{j\in \varphi [i]}\left(x-\alpha^{j}\right)$. The dimension of the BCH code is computed as $k=n-\left|Z\right|$. The minimum distance of the BCH code is at least δ based on the BCH bound [22]. A thorough theory of classic BCH codes is discussed in [21].

Steane’s enlargements of the CSS construction and Hermitian construction are widely used in quantum stabilizer codes. To proceed further, let us review some useful results as follows.

**Theorem** **1**[5,10].
*(1)* *If there exists a classical linear*${\left[n,k_{1},d_{1}\right]}_{q}$*code*C*such that*$C^{⊥E}\subseteq C$*, and*C*can be enlarged to a classical linear*${\left[n,{k^{\prime }}_{1},{d^{\prime }}_{1}\right]}_{q}$*code*$C^{\prime }$*where*${k^{\prime }}_{1}-k_{1}\geq 2$*, then there exists an*$[[n,\;{k^{\prime }}_{1}+k_{1}-n,d\geq \min$ $\{d_{1},\left\lceil \frac{q+1}{q}{d^{\prime }}_{1}\right\rceil \}]{]}_{q}$*stabilizer code;**(2)* *If there exists a classical linear*${\left[n,k_{1},d_{1}\right]}_{q^{2}}$*code*D*such that*$D^{⊥H}\subseteq D$*, then there exists an*${\left[[n,2k_{1}-n,d\geq d_{1}]\right]}_{q}$*stabilizer code.*


We construct quantum stabilizer codes using classic codes which contain their dual codes. An important lemma is generalized in [5].

**Lemma** **1**[5]. *Let*
q
*be a prime power and*
n
*be an integer such that*
$\gcd\left(n,q\right)=1$:*(1)* *A cyclic code of length*n*over*$F_{q}$*with a defining set Z contains its Euclidean dual code if and only if*$Ζ\cap {Ζ}^{-1}=\emptyset$*, where*${Ζ}^{-1}=\left\{-z\;mod\;n\;|z\in Ζ\right\}$;*(2)* *A cyclic code of length n over*$F_{q^{2}}$*with a defining set Z contains its Hermitian dual code if and only if*$Ζ\cap {Ζ}^{-q}=\emptyset$*, where*${Ζ}^{-q}=\left\{-qz\;mod\;n\;|z\in Ζ\right\}$.

## 3. Steane’s Construction

Suppose $n=r\left(q^{m}-1\right)$ and $ord_{n}\left(q\right)=2m$. If r=1, then $q^{m}\equiv 1\mathrm{mod}n$. We only consider the case where r>1.

**Lemma** **2.***If*$1<i<rq^{{}^{\left\lceil \frac{m}{2}\right\rceil }}$*,*φ[i]*has*m*elements if and only if*r|i*;*φ[i]*has*2m*elements if*r∤i .

**Proof.** If r|i , we obtain $r(q^{m}-1)|i(q^{m}-1)\Rightarrow i(q^{m}-1)\;\equiv 0\mathrm{mod}\;n\Rightarrow iq^{m}\equiv i\;\mathrm{mod}\;n$.

If m=1, we obtain iq≡imodn; therefore, φ[i] has one element. Now, let us discuss the case where m>1. Assume that φ[i] has $m_{i}$ elements, where $m_{i}|m$. If m is even, then $1<m_{i}\leq \frac{m}{2}$; if m is odd, then $1<m_{i}\leq \frac{m}{3}$. We have $iq^{m_{i}}\equiv i\;\mathrm{mod}\;n\Rightarrow n|i\left(q^{m_{i}}-1\right)\Rightarrow r\frac{q^{m}-1}{q^{m_{i}}-1}|i$. Since $1\leq i\leq rq^{\left\lceil \frac{m}{2}\right\rceil }$$<r\frac{q^{m}-1}{q^{m_{i}}-1}$, it has a contradiction. Therefore, φ[i] has m elements.

Conversely, if φ[i] has m elements, we obtain $iq^{m}\equiv i\mathrm{mod}n\Rightarrow n|i(q^{m}-1)\Rightarrow r|i$. If r∤i , assume that φ[i] has $m_{i}$ elements, where $m_{i}|m$. We have $iq^{m_{i}}\equiv i\mathrm{mod}n$. Since $r\frac{q^{m}-1}{q^{m_{i}}-1}|i$, it has a contradiction. Finally, Lemma 2 follows. □

### 3.1. m Is Even

Let us consider the case where m is even first. The following theorem contributes to choosing cyclotomic cosets.

**Lemma** **3.***If i is an integer such that*$r(q^{\frac{m}{2}}-1)|i$*, then*φ[i]=φ[−i].

**Proof.** Supposing that m is even, we have $n=r(q^{\frac{m}{2}}-1)(q^{\frac{m}{2}}+1)$. If $r(q^{\frac{m}{2}}-1)|i$, we obtain $i(q^{\frac{m}{2}}+1)\equiv 0\mathrm{mod}n\Rightarrow iq^{\frac{m}{2}}\equiv -i\mathrm{mod}n\Rightarrow \varphi [i]=\varphi [-i]$. □

According to Steane’s construction, quantum BCH codes can be generated by Euclidean dual-containing classical BCH codes, with the selected cosets in the range of $\varphi [\left(k-1\right)r(q^{\frac{m}{2}}-1)+1]\sim \varphi [kr(q^{\frac{m}{2}}-1)-1]$, $1\leq k\leq q^{\frac{m}{2}}+1$. However, some cosets are not disjointed in this range. Therefore, we should choose the cosets carefully.

**Theorem** **2.**
*Let*
q≥4
*be a prime power,*
n
*be an integer such that*
$\gcd\left(n,\;q\right)=1$
*and*
$ord_{n}(q)=2m$
*. Assume that*
$n=r\left(q^{m}-1\right)$
*, where*
1<r<q 
*. If*
m≥4
*, then there exists an*
${\left[[n,n-2m\left(2r-1\right)(q^{\frac{m}{2}}-q^{\frac{m}{2}}{}^{-1}-1)+2m,d\geq r(q^{\frac{m}{2}}-1)]\right]}_{q}$
*quantum BCH code.*


**Proof.** Since $n=r(q^{m}-1)$ and $ord_{n}(q)=2m$, we have $n|q^{2m}-1$ and $r|q^{m}+1$. If r=q−1, we obtain $q-1|q^{m}+1\Rightarrow q^{m}+1=(\sum_{i=0}^{m-1}q^{i})\left(q-1\right)+2$. Clearly, this is not true for the case where q≥4. We have 1<r<q−1.

Let $C=\langle \prod_{i}M^{\left(i\right)}(x)\rangle$ with the defining set Z, where $rq^{\frac{m}{2}-1}\leq i\leq r(q^{\frac{m}{2}}-1)-1$. If $Ζ\cap {Ζ}^{-1}\neq \emptyset$, there exist i and j such that $iq^{l}\equiv -j\mathrm{mod}n$, where $rq^{\frac{m}{2}-1}\leq i,j\leq r(q^{\frac{m}{2}}-1)-1$ and 0≤l≤2m−1. We then obtain the following:(1)$$iq^{l}+j\equiv 0\;\mathrm{mod}n$$

This congruence equation contradicts the fact that $0<iq^{l}+j\leq n-(q^{\frac{m}{2}}-1)$ when $0\leq l\leq \frac{m}{2}$. Let us consider the case where $\frac{m}{2}+1\leq l\leq m$. Thus, $0\leq m-l\leq \frac{m}{2}-1$, and it follows that $rq^{m}i+rq^{m-l}j\equiv 0\mathrm{mod}n$. Since $rq^{m}\equiv r\mathrm{mod}n$, we have $i+jq^{m-l}\equiv 0\mathrm{mod}\;(q^{m}-1)$. Since m≥4 and 1<r<q−1, we obtain $0<i+q^{m-l}j\leq r(q^{m-1}+q^{\frac{m}{2}}-q^{\frac{m}{2}-1}-1)-q^{\frac{m}{2}-1}-1\;<q^{m}-q^{m-1}+q^{\frac{m}{2}+1}-2q^{\frac{m}{2}}-q<q^{m}-1$. Therefore, the congruence equation $i+jq^{m-l}\equiv 0\;\mathrm{mod}(q^{m}-1)$ is not satisfied.

When m+1≤l≤2m−1, we have 1≤2m−l≤m−1. From $q^{2m}\equiv 1\mathrm{mod}n$, we can infer that
(2)$$i+jq^{2m-l}\equiv 0\;\mathrm{mod}n$$

Obviously, it contradicts the cases where $0\leq l\leq \frac{m}{2}$ and $\frac{m+1}{2}\leq l\leq m$. Therefore, $Ζ\cap {Ζ}^{-1}=\emptyset$, and C is Euclidean dual-containing.

Suppose φ[i]=φ[j], where $rq^{\frac{m}{2}-1}\leq i\neq j\leq r(q^{\frac{m}{2}}-1)-1$. It follows that $iq^{l}\equiv j\mathrm{mod}n$, where 1≤l≤2m−1. We thus obtain the following:(3)$$iq^{l}-j\equiv 0\mathrm{mod}n$$

When $1\leq l\leq \frac{m}{2}$, it contradicts the case where $r+1\leq iq^{l}-j\leq r(q^{m}-q^{\frac{m}{2}}-q^{\frac{m}{2}-1})-q^{\frac{m}{2}}<n$.

When $\frac{m}{2}+1\leq l\leq m$, since $rq^{m}\equiv r\mathrm{mod}n$, we have $rq^{m-l}j-ri\equiv 0\mathrm{mod}n$. Hence, $jq^{m-l}-i\equiv 0\mathrm{mod}\;(q^{m}-1)$, where $0\leq m-l\leq \frac{m}{2}-1$. If m−l=0, we have $j-i\equiv 0\mathrm{mod}(q^{m}-1)$, which contradicts the fact that $0<i\neq j<q^{m}-1$. If $1\leq m-l\leq \frac{m}{2}-1$, we have
$$r+1<jq^{m-l}-i<r(q^{m-1}-2q^{\frac{m}{2}-1})-q^{\frac{m}{2}-1}<q^{m}+q^{\frac{m}{2}-1}-q^{m-1}-2q^{\frac{m}{2}}<q^{m}-1.$$

Since 1<r<q−1 and m≥4, $jq^{m-l}-i\equiv 0\mathrm{mod}(q^{m}-1)$ is not satisfied.

When m+1≤l≤2m−1, we have 1≤2m−l≤m−1. Since $q^{2m}\equiv 1\mathrm{mod}n$, Equation (3) is transformed into $jq^{2m-l}-i\equiv 0\mathrm{mod}n$. This is similar to the cases where $1\leq l\leq \frac{m}{2}$ and $\frac{m}{2}+1\leq l\leq m$. To sum up, all the cosets given above are mutually disjointed.

From Lemma 2, there are $q^{\frac{m}{2}}-q^{\frac{m}{2}-1}-1$ cosets with m elements. Since $\varphi \left[1\right]=\varphi [q^{\frac{m}{2}}],\;\cdots ,$ $\varphi [rq^{\frac{m}{2}-1}-2]=\varphi [rq^{\frac{m}{2}}-2q]$ and $\varphi [rq^{\frac{m}{2}-1}-1]=\varphi [rq^{\frac{m}{2}}-q]$, there are $r(q^{\frac{m}{2}}-1)-1$ consecutive integers in Z. Therefore, we obtain $C={\left[n,k_{1}=n-m\left(2r-1\right)(q^{\frac{m}{2}}-q^{\frac{m}{2}-1}-1),d_{1}\geq r(q^{\frac{m}{2}}-1)\right]}_{q}$ according to the BCH bound. Let $C^{\prime }=\langle \prod_{j}M^{\left(j\right)}(x)\rangle$ and $rq^{\frac{m}{2}-1}\leq j\leq r(q^{\frac{m}{2}}-1)-2$. Since 1<r<q−1, we have $(rq^{\frac{m}{2}-1}-1)q$$\leq r(q^{\frac{m}{2}}-1)-2$, and thus $\varphi [rq^{\frac{m}{2}-1}-1]=\varphi [rq^{\frac{m}{2}}-q]$. We obtain $C^{\prime }=[n,{k^{\prime }}_{1}=n-m(2r-1)$$(q^{\frac{m}{2}}-q^{\frac{m}{2}-1}-1)+2m,{d^{\prime }}_{1}\geq r(q^{\frac{m}{2}}-1)-1{]}_{q}.$ Since ${k^{\prime }}_{1}-k_{1}=2m>2$, $C^{\prime }$ is an enlargement of *C*. Since $r(q^{\frac{m}{2}}-1)\leq \left\lceil \frac{q+1}{q}\right\rceil (r(q^{\frac{m}{2}}-1)-1)$, we have an $[[n,n-2m(2r-1)\left(q^{\frac{m}{2}}-q^{\frac{m}{2}-1}-1\right)+2m,$ $d\geq r\left(q^{\frac{m}{2}}-1\right)]{]}_{q}$ quantum BCH code. □

It is rather remarkable that q≥4 ensures that $C^{\prime }$ contains the highest numbers of consecutive integers. We choose m≥4 for the reason that there exist cyclotomic cosets φ[i]=φ[−j] when m=2. The *q*-ary cyclotomic cosets proposed in Theorem 2 not only easily compute the dimensions of C and $C^{\prime }$, but also ensure C is Euclidean dual-containing. The condition 1<r<q ensures that the selected cosets are mutually disjointed.

**Example** **1.***If*q=5*,*m=4*and*r=2*, we have n = 1248 and*$r(q^{\frac{m}{2}}-1)=48$*. It is easy to compute the following 5-ary cyclotomic cosets:*φ[10]={10,50,250,2}, …, φ[48]={48,240,1200,1008}*. Obviously,*φ[48]=−φ[48]*. Let*$C=<\prod_{i\in Z}M^{\left(i\right)}(x)>$*have the defining set*$Z=\overset{47}{\underset{i=10}{\cup }}\varphi \left[i\right]$*and*$C^{\prime }=<\prod_{j\in Z^{\prime }}M^{\left(j\right)}(x)>$*have the defining set*$Z^{\prime }=\overset{46}{\underset{j=10}{\cup }}\varphi \left[j\right]$. $C={\left[1248,1020,d_{1}\geq 48\right]}_{5}$*is Euclidean dual-containing, and*$C^{\prime }={\left[1248,1028,{d^{\prime }}_{1}\geq 47\right]}_{5}$*is an enlargement of*C*. Then, we obtain an*${\left[[1248,800,d\geq 48]\right]}_{5}$*quantum BCH code.*

### 3.2. m Is Odd

Next, we consider the case where m is odd. For simplicity, we define $Q_{1}=q^{\frac{m+1}{2}}-q+\left\lfloor \frac{r}{2}\right\rfloor$. If m=1, we have n=r(q−1) and $ord_{n}(q)=2$, which were studied in [16]. Therefore, we choose m>1 when *m* is odd. A few contributions are presented as follows.

**Theorem** **3.***Let*q*be a prime power,*n*be an integer such that*gcd(n, q)=1*and*$ord_{n}(q)=2m$*. Assume that*$n=r(q^{m}-1)$*, where*$\left\lfloor \frac{3r}{2}\right\rfloor <q\leq 2r$*. If*m>1*, then there exists an*$\left[[n,n-4m(Q_{1}-q^{\frac{m-1}{2}}\right)$ $+m(\left\lfloor \frac{Q_{1}}{r}\right\rfloor +\left\lfloor \frac{Q_{1}-1}{r}\right\rfloor -2\left\lceil \frac{q^{\frac{m-1}{2}}}{r}\right\rceil ),d\geq Q_{1}+1]{]}_{q}$*quantum BCH code.*

The proof is similar to Theorem 2.

**Example** **2.***If*q=7*,*m=3*and*n=1368*, we have*$q^{\frac{m-1}{2}}=7$*and*$Q_{1}=44$*. It is easy to compute the following 7-ary cyclotomic cosets:*φ[7]={7,49,343,1033,391,1}, …, φ[44]={44,308,788}*and*φ[45]={45,315,837,1341,1179}*. Obviously,*φ[45]=φ[−27]*. Meanwhile, the cosets which contain*φ[1], φ[2], …, φ[6]*are mutually disjointed. We choose*φ[7], φ[8] , …, φ[44]*to generate*$C={\left[1368,1170,d_{1}\geq 45\right]}_{7}$*and*φ[7], φ[8] , …, φ[43]*to generate*$C^{\prime }={\left[1368,1173,{d^{\prime }}_{1}\geq 45\right]}_{7}$*. Finally, we obtain an*${\left[[1368,975,d\geq 45]\right]}_{7}$*quantum BCH code.*

**Corollary** **1.***Let*q*be a prime power,*n*be an integer such that*gcd(n, q)=1*and*$ord_{n}(q)=2m$*. Assume that*$n=r(q^{m}-1)$*and*m>1:
*(1)* 
*If*
$r<q<\left\lfloor \frac{3r}{2}\right\rfloor$
*or*
q>2r
*, then there exists an*

${\left[[n,n-4m(q^{\frac{m+1}{2}}-q^{\frac{m-1}{2}}-q)+m(\left\lfloor \frac{q^{\frac{m+1}{2}}-q-1}{r}\right\rfloor +\left\lfloor \frac{q^{\frac{m+1}{2}}-q-2}{r}\right\rfloor -2\left\lceil \frac{q^{\frac{m-1}{2}}-1}{r}\right\rceil ),d\geq q^{\frac{m+1}{2}}-q]\right]}_{q}$
*quantum BCH code;*
*(2)* 
*If*
q<r≤2q−4
*, then there exists an*

${\left[[n,n-4m(rq^{\frac{m-1}{2}}-rq^{\frac{m-3}{2}}+\left\lfloor \frac{r}{2}\right\rfloor )+2m(q^{\frac{m-1}{2}}-q^{\frac{m-3}{2}}-1),d\geq rq^{\frac{m-1}{2}}-q+\left\lfloor \frac{r}{2}\right\rfloor +1]\right]}_{q}$
*quantum BCH code;*
*(3)* 
*If*
$2q-3\leq r\leq \frac{q^{2}+1}{2}$
*and*
m=3
*, then there exists an*

${\left[[n,n-12(rq-r-q)+3(2q-\left\lfloor \frac{q+2}{r}\right\rfloor -\left\lfloor \frac{q+1}{r}\right\rfloor -2),d\geq rq-q]\right]}_{q}$
*quantum BCH code;*
*(4)* 
*If*
$2q-3\leq r\leq q^{2}-q+1$
*and*
m=5
*, then there exists an*

${\left[[n,n-20(rq^{2}-rq-q)+5(2q^{2}-2q-\left\lfloor \frac{q+2}{r}\right\rfloor -\left\lfloor \frac{q+2}{r}\right\rfloor ),d\geq rq^{2}-q]\right]}_{q}$
*quantum BCH code;*
*(5)* 
*If*
$2q-3\leq r<q^{2}$
*and*
m≥7
*, then there exists an*

${\left[[n,n-4m(rq^{\frac{m-1}{2}}-rq^{\frac{m-3}{2}}-q)+m(2q^{\frac{m-1}{2}}-2q^{\frac{m-3}{2}}-\left\lfloor \frac{q+1}{r}\right\rfloor -\left\lfloor \frac{q+2}{r}\right\rfloor ),d\geq rq^{\frac{m-1}{2}}-q]\right]}_{q}$
*quantum BCH code.*



**Proof.** We only listed the range of *q*-ary cyclotomic cosets to generate *C* and $C^{\prime }$. The reminder proof is similar to Theorem 2.


(1)Let $C=\langle \prod {}_{i}M^{\left(i\right)}(x)\rangle$, where $q^{\frac{m-1}{2}}-1\leq i\leq q^{\frac{m+1}{2}}-q-1$. Let $C^{\prime }=\langle \prod {}_{j}M^{\left(j\right)}(x)\rangle$, where $q^{\frac{m-1}{2}}-1\leq j\leq q^{\frac{m+1}{2}}-q-2$;(2)Let $C=\langle \prod {}_{i}M^{\left(i\right)}(x)\rangle$, where $rq^{\frac{m-3}{2}}\leq i\leq rq^{\frac{m-1}{2}}-q+\left\lfloor \frac{r}{2}\right\rfloor$. Let $C^{\prime }=\langle \prod {}_{j}M^{\left(j\right)}(x)\rangle$, where $rq^{\frac{m-3}{2}}\leq j\leq rq^{\frac{m-1}{2}}-q+\left\lfloor \frac{r}{2}\right\rfloor -1$;(3)Let $C=\langle \prod {}_{i}M^{\left(i\right)}(x)\rangle$, where r−1≤i≤rq−q−1. Let $C^{\prime }=\langle \prod {}_{j}M^{\left(j\right)}(x)\rangle$, where r−1≤j≤ rq−q−2;(4)Let $C=\langle \prod {}_{i}M^{\left(i\right)}(x)\rangle$, where $rq-1\leq i\leq rq^{2}-q-1$. Let $C^{\prime }=\langle \prod {}_{j}M^{\left(j\right)}(x)\rangle$, where rq−1≤j
$\leq rq^{2}-q-2$;(5)Let $C=\langle \prod {}_{i}M^{\left(i\right)}(x)\rangle$, where $rq^{\frac{m-3}{2}}-1\leq i\leq rq^{\frac{m-1}{2}}-q-1$. Let $C^{\prime }=\langle \prod {}_{j}M^{\left(j\right)}(x)\rangle$, where $rq^{\frac{m-3}{2}}-1\leq j\leq rq^{\frac{m-1}{2}}-q-2$. □


## 4. Hermitian Construction

Let us focus on classic BCH codes over $F_{q^{2}}$. Suppose $n=r(q^{2m}-1)$ and $ord_{n}(q^{2})=2m$. We choose r>1 for the reason that we have $q^{2m}\equiv 1\mathrm{mod}n$ when r=1.

**Lemma** **4.***If*$1\leq i\leq rq^{m}$*,*φ[i]*has*m*elements if and only if*r|i*, and*φ[i]*has*2m*elements if*r∤i .

The proof is similar to Lemma 2.

**Lemma** **5.***Let*$m_{o}$*be the odd factor of*m*and*$m_{e}$*be the even factor of*m:*(1)* *If*$r\frac{q^{2m}-1}{q^{m_{o}}+1}|i$*, then*φ[i]=−qφ[i];*(2)* *If*$r\frac{q^{2m}-1}{q^{m_{e}}+1}|i$*, then*φ[i]=−qφ[qi].

**Proof.** (1) Since $n=r(q^{2m}-1)$, we have $n=r\frac{q^{2m}-1}{q^{m_{o}}+1}(q^{m_{o}}+1)$. If $r\frac{q^{2m}-1}{q^{m_{o}}+1}|i$, we have $i(q^{m_{o}}+1)\equiv 0\mathrm{mod}n\Rightarrow i\equiv -qq^{m_{o}-1}i\mathrm{mod}n$. When $m_{o}$ is odd, we obtain φ[i]=−qφ[i]. (2) Since $n=r(q^{2m}-1)$, we have $n=r\frac{q^{2m}-1}{q^{m_{e}}+1}(q^{m_{e}}+1)$. If $r\frac{q^{2m}-1}{q^{m_{e}}+1}|i$, we have $i\equiv -qq^{{}_{m_{e-2}}}qi\mathrm{mod}n$. When $m_{e}$ is even, we obtain φ[i]=−qφ[qi]. □

### 4.1. m Is Odd

**Corollary** **2.***Let*m*be an integer. If*$r(q^{m}-1)|i$*, then*φ[i]=−qφ[i].

**Theorem** **4.***Let*q*be a prime power and*n*be an integer such that*$\gcd(n,\;q^{2})=1$*and*$ord_{n}(q^{2})$ =2m*. Assume that*$n=r(q^{2m}-1)$*, where*1<r<q*. If*m>1*, then there exists an*${\left[[n,n-2m(2r-1)(q^{m}-q^{m-2}-1),d\geq r(q^{m}-1)]\right]}_{q}$*quantum BCH code.*

**Proof.** Let $D=\langle \prod {}_{i}M^{\left(i\right)}(x)\rangle$ with the defining set Z, where $rq^{m-2}\leq i$
$\leq r(q^{m}-1)-1$. If $Ζ\cap {Ζ}^{-q}\neq \emptyset$, there exist values *i* and *j* such that $iq^{2l}\equiv -qj\mathrm{mod}n$, where $rq^{m-2}\leq i,j\leq r(q^{m}-1)-1$ and 0≤2l≤4m−2. Thus, we obtain
(4)$$iq^{2l}+jq\equiv 0\mathrm{mod}n$$

First, let us consider the case where 2l=0. Equation (4) transforms into i+qj≡0modn. This contradicts the fact that $0<iq^{2l-1}+j\leq n-(q^{m}+1)<n$.

When 2≤2l≤m+1, since $\gcd(n,q^{2})=1$, Equation (4) transforms into $iq^{2l-1}+j\equiv 0\mathrm{mod}n$. This contradicts the fact that $0<iq^{2l-1}+j\leq n-(q^{m}+1)<n$.

When m+3≤2l≤2m, since $rq^{2m}\equiv r\;\mathrm{mod}\;n$, Equation (4) transforms into
(5)$$i+jq^{2m-2l+1}\equiv 0\;\mathrm{mod}\;(q^{2m}-1)$$

We obtain $i+q^{2m-2l+1}j\leq r(q^{2m-2}-q^{m-2}+q^{m}-1)-q^{m-2}-1<q^{2m}-1$, and the congruence of Equation (5) is not satisfied.

When 2m+2≤2l≤4m−2, we have 3≤4m−2l+1≤2m−1. From $q^{4m}\equiv 1\;\mathrm{mod}\;n$, it can be inferred that $i+jq^{4m-2l+1}\equiv 0\;\mathrm{mod}\;n$. Obviously, this contradicts the cases where 0≤2l≤m+1 and m+3≤2l≤2m. Therefore, $Ζ\cap {Ζ}^{-q}=\emptyset$, and *D* is Hermitian dual-containing.

Similar to Theorem 2, the cosets $\varphi [rq^{m-2}]$, …, $\varphi [r(q^{m}-1)-2]$ and $\varphi [r(q^{m}-1)-1]$ are mutually disjointed. From Lemma 4, there are $q^{m}-q^{m-2}+1$ cosets with m elements. Since $\varphi [1]=\varphi [q^{m-1}]$, …, $\varphi [rq^{m-2}-2]=\varphi [rq^{m}-2q^{2}]$ and $\varphi [rq^{m-2}-1]=\varphi [rq^{m}-q^{2}]$, there are $r(q^{m}-1)-1$ consecutive integers in Z. Therefore, we obtain $D={\left[n,n-m(2r-1)(q^{m}-q^{m-2}-1),d_{1}\geq r(q^{m}-1)\right]}_{q^{2}}$. Then, an ${\left[[n,n-2m(2r-1)(q^{m}-q^{m-2}-1),d\geq r(q^{m}-1)]\right]}_{q}$ quantum BCH code can be obtained by a Hermitian construction. □

It is rather remarkable that the *q*^2^–ary cyclotomic cosets proposed in Theorem 4 can easily compute the dimensions of the BCH codes. The condition 1<r<q ensures that the selected cosets are mutually disjointed. Furthermore, the cosets contain the highest numbers of consecutive integers.

**Theorem** **5.**
*Let*
q
*be a prime power,*
n
*be an integer such that*
gcd(n,q)=1
*and*
$ord_{n}(q^{2})=2m$
*, Assume that*
$n=r(q^{2m}-1)$
*, where*
q<r<2q
*. If*
m>1
*, then there exists an*
[[n,n−4m
$(q^{m+1}-q^{m-1}+r)+2m(\left\lfloor \frac{q^{m+1}-1}{r}\right\rfloor -\left\lfloor \frac{q^{m-1}+1}{r}\right\rfloor +4),$
$d\geq q^{m+1}+r)]{]}_{q}$
*quantum BCH code.*


The proof is similar to Theorem 4.

**Example** **3.***If*q=7*,*m=3*and*n=1176480*, it is easy to compute the following 49-ary cosets:*φ[50]={50,2450,120050}, …, φ[2410]={2410,118090,1080490}*. Let*$D=<\prod_{i=50}^{2410}M^{\left(i\right)}(x)>$*, where*$D={\left[1176480,1163025,d_{1}\geq 2411\right]}_{49}$*is Hermitian dual-containing. Then, we obtain an*${\left[[1176480,1149570,d\geq 2411]\right]}_{7}$*quantum BCH code.*

### 4.2. m Is Even

Now, we consider the case where m is even. A few contributions are presented as follows.

**Corollary** **3.***When letting*λ*be an integer such that*$0\leq \lambda \leq r\frac{\left(q^{m}-1)(q-1\right)}{q^{m-1}+1}$*, we have*$\varphi [r(q^{m}-1)+\lambda q^{m-1}]=$ $\varphi [-q(rq(q^{m}-1)-\lambda )]$*. In particular, when letting*i*be an integer such that*$r(q^{m}-1)|i$*, we have*φ[i]=−qφ[qi].

**Proof.** Since $rq^{2m}\equiv r\mathrm{mod}n$, we have
$$(r(q^{m}-1)+\lambda q^{m-1})q^{m}\equiv rq^{2m}-rq^{3m}+\lambda q^{2m-1}\equiv -q(rq(q^{m}-1)-\lambda )q^{2m-2}\mathrm{mod}n$$

If m is even, clearly, we obtain $\varphi [r(q^{m}-1)+\lambda q^{m-1}]=\varphi [-q(rq(q^{m}-1)-\lambda )]$. In the condition of $rq(q^{m}-1)-\lambda \geq r(q^{m}-1)+\lambda q^{m-1}$, we obtain $0\leq \lambda \leq r\frac{\left(q^{m}-1)(q-1\right)}{q^{m-1}+1}$. According to Corollary 3, if $r(q^{m}-1)|i$, we have φ[i]=−qφ[qi]. □

Therefore, we should choose the *q*^2^-ary cyclotomic cosets properly to ensure the cyclic code is Hermitian dual-containing.

**Theorem** **6.***Let*q*be a prime power,*n*be an integer such that*$\gcd(n,\;q^{2})=1$*and*$ord_{n}(q^{2})=2m$*. Assume that*$n=r(q^{2m}-1)$*, where*1<r<q*. Then, there exists an*$[[n,n-2m(2r-1)(q^{m}-q^{m-2}),$ $d\geq rq^{m}+1]{]}_{q}$*quantum BCH code.*

The proof is similar to Theorem 4.

**Example** **4.**
*If*
q=3
*,*
m=4
*,*
r=2
*and*
n=13120
*, we choose*
φ[19]
*to*
φ[162]
*as the 9-ary cyclotomic cosets, which are mutually disjointed, to generate D. Obviously,*
$Ζ\cap {Ζ}^{-3}=\emptyset$
*, where Z is the defining set of D. Then,*
$D={\left[13120,12256,d\geq 163\right]}_{9}$
*is Hermitian dual-containing. Thus, we can construct an*
${\left[[13120,11392,d\geq 163]\right]}_{3}$
*quantum BCH code.*


## 5. Comparison and Conclusions

In this section, we give some comparisons to corresponding references.

Aly et al. constructed quantum BCH codes over $F_{q}$ with classic non-primitive narrow-sense BCH codes and $F_{q^{2}}$ with classic primitive narrow-sense BCH codes [5]. In this paper, we designed quantum BCH codes with classic non-primitive, non-narrow-sense BCH codes. In [5], Aly et al. designed an ${\left[[n,n-4m\left\lceil \left(\delta -1)(1-1/q\right)\right\rceil ,d\geq \delta ]\right]}_{q}$ quantum BCH code, where $2\leq \delta \leq \delta_{\max}\leq r\frac{q^{m}-1}{q^{m}+1}<r$. If $r\frac{q^{m}-1}{q^{m}+1}<2$, a quantum BCH code does not exist. Therefore, we could not obtain quantum codes with r=2 in [5]. In this paper, we designed quantum BCH codes without this restriction. For example, if q=5, m=4 and r=2, we can construct an ${\left[[1248,\;800,\;d\geq 48]\right]}_{5}$ quantum BCH code, in which $\delta_{\max}=1.933<2$. Since $2\leq \delta \leq \delta_{\max}<r$, we got better lower bounds for the minimum distances than those in [5]. Meanwhile, [10] presented similar results to [5] with Steane’s construction. Therefore, our results were better than those in [10], too. Table 1 shows more precise conclusions.

In [17], by letting $n=r(q^{3}-1)$ and $m=ord_{n}(q^{2})=3$, quantum BCH codes were constructed with classic non-primitive, narrow-sense and non-narrow-sense BCH codes. However, in [17], quantum BCH codes could only be constructed with a fixed length n for q=3l+2. In this paper, we extended the construction to a larger range of n over any finite field $F_{q}$.

In [23], non-binary primitive quantum BCH codes were constructed when $m=ord_{n}(q)=2$ and $m=ord_{n}(q^{2})=2$. In this paper, we designed nonbinary, non-primitive quantum BCH codes. Moreover, we extended the results to more general cases where m>3.

The earlier work of this paper was conducted in [20]. In [20], we discussed the construction of quantum BCH codes with multiplicative order m=2 when the code lengths were $n=r\left(q+1\right)$ over $F_{q}$ and $\;n=r\left(q^{2}+1\right)$ over $F_{q^{2}}$. We also considered the situation where m=3 and when the code lengths were $n=r\left(q-1\right)$ over $F_{q}$ and $n=r(q^{2}-1)$ over $F_{q^{2}}$. In this paper, we discussed more general cases. We enlarged the multiplicative order to any even integers. Moreover, we extended the construction to a larger range of code lengths with $n=r(q^{m}-1)$ over $F_{q}$ and $n=r(q^{2m}-1)$ over $F_{q^{2}}$ where m denotes the integers.

In this paper, a new family of quantum BCH codes was constructed by Steane’s construction and Hermitian construction. By studying the suitable cyclotomic cosets, we tried to find such classic BCH codes which contained their dual codes. Then, we constructed nonbinary quantum BCH codes with given parameter sets. We have proven that the cyclotomic cosets are good tools to study quantum BCH codes. The defining sets contained the highest numbers of consecutive integers. Compared with the results in the references, the new quantum BCH codes had better code parameters without restrictions and better lower bounds for the minimum distances. What is more, the new quantum codes can be constructed over any finite fields, which enlarges the range of quantum BCH codes.

## Figures and Tables

**Table 1 entropy-23-00712-t001:** Code comparison with length $n=r(q^{m}-1)$.

New Quantum BCH Codes	Quantum BCH Codes in [5,10]
$${\left[[315,195,d\geq 16]\right]}_{4}$$	$${\left[[315,279,d\geq 4]\right]}_{4}$$
$${\left[[1248,800,d\geq 48]\right]}_{5}$$	—
$${\left[[1368,975,d\geq 45]\right]}_{7}$$	$${\left[[1368,1344,d\geq 3]\right]}_{7}$$
$${\left[[1533,1158,d\geq 56]\right]}_{8}$$	$${\left[[1533,1521,d\geq 2]\right]}_{8}$$
$${\left[[2736,2142,d\geq 54]\right]}_{7}$$	$${\left[[2736,2664,d\geq 7]\right]}_{7}$$
$${\left[[4599,3831,d\geq 69]\right]}_{8}$$	$${\left[[4599,4515,d\geq 8]\right]}_{8}$$
$${\left[[4800,3824,d\geq 96]\right]}_{7}$$	—

## Data Availability

Not applicable.

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
