# Peer review of "Some New Quantum BCH Codes over Finite Fields"

_entropy, 2021, doi:10.3390/e23060712_

Round 1
Reviewer 1 Report
In this paper the authors present new constructions for non-binary quantum BCH codes, extending previous constructions. Some example codes are given illustrating the advantages of the new constructions.
I recommend that the authors refer to the paper: Zhang, Ming & Li, Zhuo & Xing, Lijuan & Tang, Nianqi. (2019). Some Families of Quantum BCH Codes. International Journal of Theoretical Physics. 58. 10.1007/s10773-018-3959-0 (two of the authors are the same as for the current paper) and clearly indicate how the work in that paper is extended in the present one.
On page 2, in lines 71 and 74, I am not entirely convinced that the set Z is the same in both instances. Please clarify so as to avoid confusion.
There is a typo in line 91 on page 3 – “coeds”, should be “codes”.
There are a number of grammatical errors and therefore I would suggest that the paper is proofread. Some of this include:
Page 1, Line 17: (and in many other instances) – “most numbers of consecutive integers” – should be “highest number of consecutive integers”.
Page 1, Line 17: “Compare with the results in…” – replaced by “Compared with the results in …”
Page 5, Line177: “prove” should be “proof”
Reviewer 2 Report
The manuscript concerns the valid problems related to the protocols applied in quantum communication processes – quantum error correcting codes (QECC). In particular, the Authors propose a method of construction of such codes that uses Bose-Chaudhuri-Hocquenghem (BCH) codes over finite fields. Since quantum stabilizer codes are usually constructed with an application of classical error correcting codes, they attempt to find such classical BCH codes by studying the suitable cyclotomic cosets. After presenting a proposal of the construction of nonbinary quantum BCH codes, the Authors show how a new family of such codes can be realized by nonbinary Calderbank-Shor-Steane construction with an application of Steane enlargement. They prove that the cyclotomic cosets are good tools in studying quantum BCH codes. After comparing the results presented in the manuscript to those given in the cited references, they show that the new quantum BCH codes exhibit more favorable characteristics than those already discussed in the literature. Additionally, the proposed quantum codes can be constructed over any finite fields enlarging the range of possible BCH codes.
The manuscript is of the mathematical character. It is well written and is accessible for Readers with some mathematical background. The ideas presented by the Authors are novel and seem to be valid enough to be published. Therefore, I recommend the article for publication in its present form. I would like to point out that the references’ numbering in the main body of the paper should be reordered.
